# Inhibiting the copper efflux system in microbes as a novel approach for developing antibiotics

Aviv Meir[1], Veronica Lepechkin-Zilbermintz[1], Shirin Kahremany[2],
Fabian Schwerdtfeger[1,3], Lada Gevorkyan-Airapetov[1], Anna Munder[1], Olga Viskind[1],
Arie Gruzman[1] *, Sharon Ruthstein[1] *

1 Chemistry Department, Faculty of Exact Sciences, Bar Ilan University, Ramat-Gan, Israel, 2 Gavin Herbert Eye Institute and the Department of Ophthalmology, University of California, Irvine, California, United States of America, 3 Faculty of Biology, Albert-Ludwigs-University Freiburg, Centre for Biological Signaling Studies (BIOSS), Freiburg, Germany

☯ These authors contributed equally to this work.
* Sharon.ruthstein@biu.ac.il (SR); gruzmaa@biu.ac.il (AG)

**Data Availability Statement:** All relevant data are within the manuscript and its Supporting Information files.

## Abstract

Five out of six people receive at least one antibiotic prescription per year. However, the ever-expanding use of antibiotics in medicine, agriculture, and food production has accelerated the evolution of antibiotic-resistant bacteria, which, in turn, made the development of novel antibiotics based on new molecular targets a priority in medicinal chemistry. One way of possibly combatting resistant bacterial infections is by inhibiting the copper transporters in prokaryotic cells. Copper is a key element within all living cells, but it can be toxic in excess. Both eukaryotic and prokaryotic cells have developed distinct copper regulation systems to prevent its toxicity. Therefore, selectively targeting the prokaryotic copper regulation system might be an initial step in developing next-generation antibiotics. One such system is the Gram-negative bacterial CusCFBA efflux system. CusB is a key protein in this system and was previously reported to play an important role in opening the channel for efflux *via* significant structural changes upon copper binding while also controlling the assembly and disassembly process of the entire channel. In this study, we aimed to develop novel peptide copper channel blockers, designed by *in silico* calculations based on the structure of CusB. Using a combination of magnetic resonance spectroscopy and various biochemical methods, we found a lead peptide that promotes copper-induced cell toxicity. Targeting copper transport in bacteria has not yet been pursued as an antibiotic mechanism of action. Thus, our study lays the foundation for discovering novel antibiotics.

## Introduction

Antibiotic resistance in pathogenic microbes has become one of the most serious threats to global health in recent years. There is no doubt that we already have entered the "inactive antibiotics" era, with dozens of multi-drug resistant bacteria. For example, β-lactam antibiotics,

**Funding:** SR acknowledges the support of ISF grant no. 176/16

**Competing interests:** The authors have declared that no competing interests exist.

which are the most common class used today, were the first medications developed to effectively treat bacterial infections, and are credited with having saved innumerable lives [1–4]. However, an unfortunate consequence of using these life-saving antibiotics has been a steep rise in bacterial strains resistant to treatment. Many cases of penicillin-resistant bacterial infections, as well as resistance to second- and third-generation antibiotics have been reported in recent years [5, 6]. Thus, developing effective alternatives to treat bacteriogenic diseases is an urgent clinical need [7–11].

One method of bactericidal intervention may be to interrupt copper transfer through bacterial membranes. Specific inhibition of components unique to bacterial transport systems could lead to bacterial death due to intracellular accumulation of copper, which would accelerate the formation of reactive oxygen species and enhance the Fenton reaction, thereby killing microbes [12]. Prokaryotic cells have four main mechanisms for Cu(I) transport [13]: (i) the Cytoplasmic CueR-metal sensor, which initiates the transcription process of CueO and CopA upon Cu(I) binding [14], (ii) CopA transfer, wherein the ATPase protein relies on the energy of ATP hydrolysis to actively transfer Cu(I) ions from the cytoplasm to the periplasm [15], (iii) a CusF metallochaperone, which escorts Cu(I) ions from CopA to the CusCBA efflux system, which transports Cu(I) from the periplasm to the extracellular domain [16], and (iv) CueO, which oxidases Cu(I) to less toxic Cu(II) [17]. Targeting CusCFBA and CueR systems for antibiotics research has a fundamental advantage because they are only found in prokaryotic cells, and no homologs exist in eukaryotic cells. Thus, targeting one of these two copper transport systems as antibiotics should operate solely on bacteria and would not interfere with the human copper transport mechanism.

CueR is a metal sensor that senses the Cu(I) ion with high affinity and induces a transcription process [14, 18]. It is found in a large variety of microbes, but their structure and functionality vary among bacterial species, meaning that a single compound will be applicable for a broad range of bacteria [19, 20]. By contrast, CusCFBA efflux systems are less ubiquitous, but are found in many pathogenic microbes such as *Legionella*, *Salmonella*, *Klebsiella*, *Pseudomonas* and many other Gram-negative bacteria. The sequence identity among all species is about 30%.

The CusCFBA complex consists of an inner membrane proton-substrate carrier (CusA) and an outer membrane protein (CusC). The inner and outer membrane proteins are connected by a linker protein CusB in the periplasm, and are at an oligomerization ratio CusA: CusB:CusC of 3:6:3 which form a channel [21]. The structure of the entire channel has not been resolved; however, the crystal structures of the individual components (CusC, CusA, and CusB) and of the CusBA complex have been determined [22–25]. It has been suggested that the CusB-CusF interaction functions as the trigger for the entire CusCFBA efflux system opening, galvanizing the transfer of Cu(I) to CusC [26, 27]. The crystal structure of CusB indicates that the protein is folded into an elongated narrow structure and can be separated into four domains. The first three domains are mostly β-strands, whereas the fourth forms a three-helix bundle [22, 24]. The most studied region in CusB is the N-terminal domain (CusBNT), which comprises 61 amino acids (residues 28–88) that were shown to interact directly with CusF [28, 29]. CusBNT knockout studies have resulted in inhibition of cell growth [30, 31]. Molecular dynamics (MD) simulations on CusBNT show structural disorders in both the apo and holo forms of CusBNT, including a slight local stabilization around the Cu(I) binding site [32]. Using electron paramagnetic resonance (EPR) spectroscopy-based nanometer distance measurements, we have previously shown that CusB undergoes major structural changes upon Cu (I) binding, mainly at the two centered domains [33]. Our results are in line with previous gel filtration chromatography experiments that suggest a more compact structure upon Cu(I) binding [31]. We reported two model structures of the CusB dimer in solution: apo-CusB and

holo-CusB (Fig A in S1 File) [31]. Moreover, it was found that under copper stress, CusB changes its conformation and shifts the equilibrium from a disassembled CusCBA complex to an assembled complex, stressing its significance not only as a linker and Cu(I) binding protein, but also in playing a significant role in assembling the channel for proper Cu(I) transport [31, 34]. Using computational modeling, we have herein designed peptides that might bind CusB to two central domains and inhibit the structural changes formed upon Cu(I) stress. Such an effect may reduce bacterial viability. We found that a leading peptide (pep **5**) reduces *E. coli* viability in a high copper concentration environment. Using EPR spectroscopy, we observed a loss of CusB structural changes in the presence of pep **5.** Taken together, we can conclude that pep **5** could be used as a structural template for developing novel antibiotic candidates with a unique mechanism of action not targeted by intellectual drug design.

## Materials and methods

### Protein preparation and peptide library generation

The crystal structure of CusB (PBD code: 3H94) was subjected to the protein preparation wizard, as implemented in Schrödinger software. Default settings were used. Missing hydrogens were added, the protonation states of residues were optimized, and limited minimization (up to an RMSD of 0.3 Å with respect to the crystal structure) was carried out. Peptides were built using Discovery Studio's "build and edit protein" protocol [35] and prepared with Schrödinger's Ligprep [36].

### Binding pocket and docking

Potential active sites for peptides on CusB were identified using Schrödinger's SiteMap. Site-Map characterizes binding sites based on various properties such as size, volume, and amino acid exposure. Default parameters were used. Three binding sites were identified. Peptides were docked into the large site of the crystal structure of CusB, which is in agreement with our hypothesis for the inhibition area site, using Glide XP protocol [37, 38], as implemented in Schrodinger. Default settings were used. The docking box was centered on the identified sequence and the resulting poses were sorted on the basis of their XP gscores.

### Pharmacophore mapping

Pharmacophore model was build using Phase as implemented in Schrödinger software; it was based on Glide XP scoring terms for fragments docked to the Workspace receptor. In this method, the features are chosen to maximize the binding, but instead of a single ligand, fragments with the features are docked, and common features are chosen that satisfy criteria on their positions and directions. Clustering is performed to reduce the number of fragments [39, 40].

The program is used the following calculations: H-bond donors and acceptors as directed vectors, positive and negative ionizable areas, and finally, lipophilic fragments represented by spheres. Excluded volume spheres are also included in the model, based on coordinates defined by amino acid side chain atoms in order to depict the inaccessible areas for any potential ligand. This pharmacophore model was used as a query in the virtual screening of the peptide library.

### Peptide synthesis

The peptides were synthesized using rink-amide resin (GLC, Shanghai, China). Couplings of standard Fmoc (9-fluorenylmethoxy-carbonyl)-protected amino acids were achieved with

(O-Benzotriazol-1-yl)-N,N,N′,N′-tetramethyluronium (HBTU, Bio-Lab Ltd, Jerusalem, Israel) in N,N-dimethylformamide (DMF, Bio-Lab Ltd, Jerusalem, Israel) combined with N,N-Diiso-propylethylamine (DIPEA, Merck, Rehovot, Israel) for a 1 h cycle. Fmoc deprotection was achieved with 20% piperidine/DMF solution (Alfa Aesar, Ward Hill, MA, USA). Ninhydrine tests were performed after each coupling set. Side-chain deprotection and peptide cleavage from the resin were achieved by treating the resin-bound peptides with a 5 mL cocktail of 95% trifluoroacetic acid (TFA, Bio-Lab Ltd, Jerusalem, Israel), 2.5% triisopropylsilane (TIS, Merck, Rehovot, Israel), and 2.5% $H_2O$ for 4.5 h. The peptides were washed four times with cold diethyl ether, vortexed, and then centrifuged for 10 min at 3500 rpm. The mass of the peptide was confirmed either by a MALDI-TOF MS-Autoflex III-TOF/TOF mass spectrometer (Bruker, Bermen, Germany) equipped with a 337 nm nitrogen laser or with ESI (electron spray ionization) mass spectrometry on a Q-TOF (quadruple time-of-flight) high-resolution micromass spectrometer (Agilent, Santa Clara,CA, USA) (Figs B to D in S1 File). Peptide samples were typically mixed with two volumes of premade dihydrobenzoic acid (DHB) matrix solution, deposited onto stainless steel target surfaces, and allowed to dry at room temperature.

## CusB cloning, expression, and purification

CusB gene was isolated from *E. coli* genomic DNA by PCR using primers containing specific CusB sequences and flanking regions that correspond to the expression vector sequences of pYTB12 (5′ primer- GTTGTACAGAATGCTGGTCATATGAAAAAAATCGCGCTTATTATCG and 3′ primer- GTCACCCGGGCTCGAGGAATTTCAATGCGCATGGGTAGC). This amplicon was cloned into the pYTB12 vector using the free ligation PCR technique [41]. This construct, which encodes for the fusion protein composed of CusB, an intein, and a chitin-binding domain, was transformed into the *E. coli* strain BL21 (DE3). The CusB construct was expressed in BL21 cells, which were grown to an optical density of 0.6–0.8 at 600 nm and were induced with 1 mM isopropyl-β-D-thiogalactopyranoside (CALBIOCHEM) for 20 h at 18˚C. *E. coli* were harvested by centrifugation, and the pellets were subjected to three freeze–thaw cycles. Pellets were resuspended in lysis buffer (25 mM $Na_2HPO_4$, 150 mM NaCl, and 200 μM PMSF; pH 7.5). *E. coli* were sonicated by 12 bursts of 30 s, each with a 30 s cooling period between bursts (65% amplitude). After sonication, the cells were centrifuged, and the soluble fraction of the lysate was passed through a chitin bead column (New England Biolabs, Ipswich, MA, USA), allowing the CusB fusion to bind to the resin via its chitin-binding domain. Resin was then washed with 30 column volumes of lysis buffer. To induce intein-mediated cleavage, beads were incubated in 50 mM dithiothreitol (DTT), 25 mM $NaH_2PO_4$, and 150 mM NaCl at pH 8.9, for 40 h at room temperature. CusB was collected in elution fractions and analyzed using silver-stained SDS PAGE (10% glycine) [42]. Mutant formation was carried out using an identical protocol.

## CusB spin labeling

Before labeling, 10 mM DTT was added to the protein solution and mixed overnight at 4˚C. DTT was dialyzed out using a Microsep Advanced Centrifugal Device (Pall, Port Washington, NY, USA) applied to samples of up to 5 mL in lysis buffer, with a 3-kDa molecular weight cut-off. Next, 0.25 mg of *S*-(2,2,5,5-tetramethyl-2,5-dihydro-1H-pyrrol-3-yl)methyl methanesulfo-nothioate (MTSSL, TRC) was dissolved in 15 μL Dimethyl sulfoxide (DMSO) and added to 0.75 ml of 0.01 mM protein solution (20-fold molar excess of MTSSL). The protein solution was then vortexed overnight at 4˚C. The free spin label and Cu(I) ions were removed by several dialysis cycles over 4 days. A sample of the running-washing buffer was taken for Continuous-Wave (CW) EPR measurement at room temperature (RT), and no free-spin EPR signal was

observed. The concentration was determined with a Lowry assay [43]. The final concentration of spin-labeled CusB protein was 0.01–0.02 mM.

## Addition of Cu(I) ion to protein solution

Tetrakis (acetonitrile) copper(I) hexafluorophosphate (Sigma-Aldrich, St. Louis, MO, USA) was added to the protein solution under nitrogen gas to maintain inert anaerobic conditions. No Cu(II) EPR signal was observed at any time. A Cu(I):CusB ratio of 3:1 was used for all EPR measurements.

## Electron paramagnetic resonance (EPR) spectroscopy

A constant-time four-pulse double electron-electron resonance (DEER) experiment with pulse sequence $\pi/2(\nu_{obs})$-$\tau_1$-$\pi(\nu_{obs})$-$t'$−$\pi(\nu_{pump})$-$(\tau_1 + \tau_2 - t')$-$\pi(\nu_{obs})$-$\tau_2(\nu_{obs})$-$\tau_2$-echo was performed at ($50 \pm 0.5$ K) on a Q-band Elexsys E580 equipped with a 2-mm probe head; bandwidth, 220 MHz. A two-step phase cycle was applied to the first pulse. The echo was measured as a function of $t'$, and $\tau_2$ was kept constant to eliminate relaxation effects. The pump pulse frequency was set to the maximum of the EPR spectrum and the observer pulse frequency was set 60 MHz higher than that of the pump pulse. The observer $\pi/2$ and $\pi$ pulses, as well as the $\pi$ pump pulse, had durations of 40 ns; the dwell time was 10 ns. The observed frequency was 33.82 GHz. The parameter $\tau_1$ was set to 200 ns, and $\tau_2$ was set to 1200 ns. The repetition time was set to 12 ms, and 30 shots per point were applied. The samples were measured in 1.6-mm quartz capillary tubes (Wilmad-Labglass, Vineland, NJ, USA). The data were analyzed with the Deer-Analysis 2016 program and with Tikhonov regularization [44, 45]. The regularization parameter in the L curve was optimized by examining the fit of the time-domain data and was found to be between 30 and 40.

## *E. coli* growth rate experiments

*E. coli* (BL21;DE3) growth rates were evaluated in 96-well microplates: in 200 µl of a poor medium comprising M9 medium. Incubation was for 16 h at 37˚C with Copper(II) chloride (Sigma-Aldrich, St. Louis, MO, USA) at a concentration of 5 µM. Absorbance at 600 nm was measured every 30 min after stirring for 5 s in a 96-well plate reader (SPL Life Sciences, Gyeonggi, Korea). The buffer background was auto-subtracted. Each experiment was repeated five times under identical conditions.

## *E. coli* viability assay

*E. coli* (BL21;DE3) were dyed using the L7007 LIVE/DEAD Bacterial Viability kit (Molecular Probes, Eugene OR, USA); microbes were grown in M9 medium. Images were acquired on a Leica SP8 confocal microscope, running LASX acquisition software. The magnification consisted of a 63 X 1.4 NA objective. Microbes were counted using ImageJ software. Error calculations were based on three repetitions, with each experiment scanned at nine different tiles—all together, 27 repetitions.

## Rat L6 myotubes culture

The cells were maintained as previously described [46]. All experiments were conducted on fully differentiated myotubes. Compound **8** was used as a positive cytotoxic for L8 myotubes control at concertation 100 µM [47].

## Determination of the purity of test peptides by analytical HPLC

The peptides were purified by preparative reversed phase HPLC (Luna 5μm C18; 100 A; 250 mm X 4.6 mm, Phenomenex Torrance, CA, USA) and analyzed by identical system using analytical column.

## Results

### CusB inhibitor design

CusB (PDB code: 3H94) was found to be a dimer in solution, which assembles to a homo hexamer in the whole CusCBA channel [13, 23, 24, 33]. The workflow for finding a potential inhibitor of CusB was as follows: (i) structure preparation and library preparation, (ii) binding site detection, (iii) pharmacophore feature definition, and (iv) docking of the top-ranked peptides obtained from pharmacophore screening. A short sequence from CusB ([236]AKIQGMDPVWVTA[248]) was selected as a target sequence for mimicry, and a short peptide of identical sequence was made. This area is responsible for the flexibility of CusB domains 2 and 3, which has been found to be essential for opening the CusCBA channel.[33, 48] Moreover, the assembly of the CusB within the CusCBA transporter was found to be a key part in the function of this protein.[34] We suspected that this short peptide would interfere with the interaction between two monomers of CusB, thereby impeding the ability of the channel to be physiologically active. However, this 13-amino acid peptide was too long and contained too many pharmacophore features. Thus, several overlapping, short peptides were designed, based on the entire peptide sequence. To more empirically confirm our choice of a potential inhibition area, we also identified potential binding sites for peptide binding using Schrodinger's SiteMap [36] (Fig 1).

Based on these results, we defined the cavity area and designed a structure-based pharmacophore model by using the receptor cavity protocol implemented in Schrodinger's Phase [39] (Fig 2).

The resulting pharmacophores were further refined by adding excluded volumes. Excluded volumes are points in space occupied by protein atoms, which represent steric hindrances in

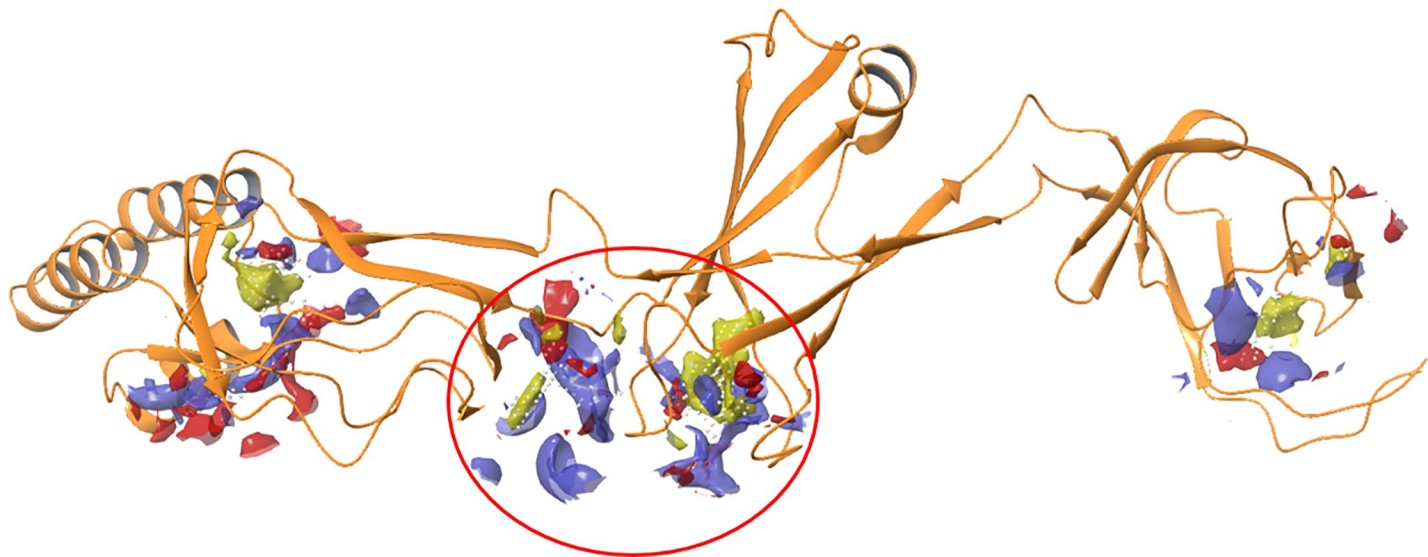

**Fig 1. Representation of the CusB (PBD code: 3H94) [236]AKIQGMDPVWVTA[248] sequence overlapping with the binding site, circled in red, calculated with the Schrödinger Sitemap program.**

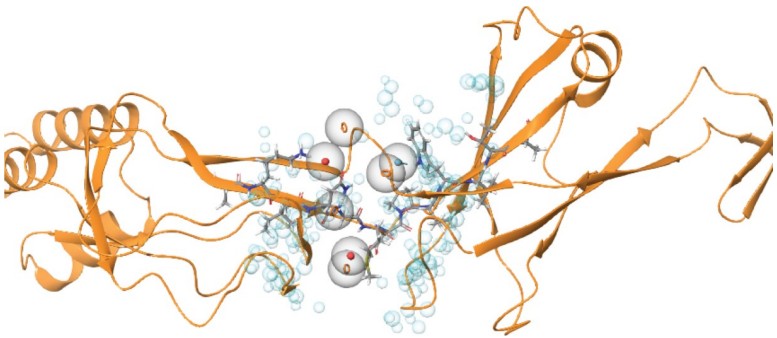

**Fig 2.** CusB-based pharmacophore hypothesis (red sphere/circle: Negative region acceptor, orange ring: Aromatic ring, light-blue: Hydrogen bond donors and blue sphere: Excluded volume).

the binding site. Such points cannot be occupied by ligand atoms. This pharmacophore model was used for virtual screening of a peptide library. These peptides were subjected to a conformational search procedure prior to the screening. The conformations were subsequently fit to the pharmacophore models using flexible fitting methods. The twenty highest ranked peptides, based on fit value, were chosen for validation with docking simulations. The hit molecules, determined by the virtual screening, were refined [39] to determine whether these chemical features were mapped with structure-based interaction mode. Ten peptides, sorted and chosen on the basis of XP gscore, were chosen for synthesis and further evaluation (Table 1). Nine chosen peptides were short subsequences of the parent peptide (pep1).

The peptides were synthesized using Rink resin. The purity level of some of the peptides was below 95% (Table 1), despite several attempts to purify them. However, the preliminary screening of the peptides took into account the option that if some of the low purified peptides showed good activity, additional synthesis will be conducted.

### Effect of test peptides on *E. coli* growth

The aim of the copper stress assay was to identify those peptides most effective at killing or at suppressing the growth of *E. coli* by inhibition of CusB protein functioning. We hypothesized that peptides would penetrate into *E. coli* and bind to CusB, interfere with its physiological dimerization, and thereby induce non-physiological structural changes or inhibit physiological structural changes in the transporter in response to extended levels of Cu(I). Thus, *E. coli* will lose its ability to be protected by copper-mediated oxidative stress, resulting in the death of the

**Table 1. The peptides used in this study.**

| Peptide | Sequence | MW (g/mole) | Purity (%) |
|---|---|---|---|
| pep1 | AKIQGMDPVWVTA | 1413.74 | 96.0 |
| pep2 | KIQGMDPVWVT | 1271.67 | 96.9 |
| pep3 | AKIQGM | 646.35 | 84.6 |
| pep4 | DPVWVTA | 785.41 | 92.1 |
| pep5 | IQGMD | 598.26 | 94.2 |
| pep6 | GMDPVW | 702.32 | 92.3 |
| pep7 | VWVTA | 573.33 | 97.4 |
| pep8 | AKIQG | 514.32 | 97.4 |
| pep9 | MDPV | 459.22 | 96.4 |
| pep10 | PVWV | 498.30 | 75.4 |

bacteria. Unlike CusB, CueO has homologs in human cells (Superoxide dismutase, SOD), and the main role in this study is to affect the CusCFBA efflux system regardless to CueO function, all measurements were carried out with an active CueO. First, we verified whether all ten peptides were stable under physiological conditions *via* incubation at 37˚C for 24 h by ESI-MS analysis (Figs B to D in S1 File). Peptides 2–10 were stable under these given conditions, whereas peptide 1 was unstable.

Next, *E. coli* were grown in M9 medium for 16 h, and their growth rate was evaluated in the absence and presence of Cu(II) in the medium. In each sample, a different peptide was introduced to the medium, with the kanamycin as a positive control. Though CusB binds only to Cu(I) and not to Cu(II), previous studies have demonstrated the ability of Cu(II) to penetrate into the cells and then participate in the copper cycle after being reduced to Cu(I) [49–51]. The tissue culture experiments were performed under anaerobic conditions. Growth rate values for all peptides at various concentrations after 14 h of incubation are presented in Fig 3 (the growth rate curves are presented in Figs E to F in S1 File). A comparison of growth rates among the various peptides revealed substantial differences in the ability of *E. coli* to grow in the presence of the peptides, especially upon the addition of Cu(II) into the growth media. Pep5 exhibited the most significant decrease in O.D. from 54.5 ± 4.2% at a 50 μM concentration to 77.2 ± 6.4% decrease at a 100 μM concentration. Pep8 also exhibited an impressive effect: the O.D. value decreased until 45.5 ± 4.8% in the treatment at a 50 μM concentration and to 63.6 ± 4.0% at a100 μM concentration. At a 125 μM peptide concentration, all peptides exhibited a major reduction in O.D. both in the absence and presence of Cu(II).

## Effect of test peptides on *E.coli* viability

Live/dead fluorescence cell imaging experiments were used to indicate bacterial viability when incubated with the peptides. This method also helped to determine whether the decrease in the growth rate described above resulted from deceleration in *E. coli* growth or its substantial death. *E coli* were grown to the late lag phase (according to the growth rate lines; Figs B to D in S1 File) in the presence of the peptides in poor medium (M9), which mostly resembles physiological conditions.[52–54] Glucose levels in the growth media were sufficient to prevent bacterial viability loss due to lack of nutrition, yet not too high to positively affect its viability [55, 56]. A comparison of cell counts based on the images presented in Fig 4 (additional images are presented in Figs G to K in S1 File) revealed a high correlation between the cytotoxic effect caused by the peptides *via* Cu(II) homeostasis interference and *E. coli* death. Though a slight reduction in bacteria viability was observed in the control sample, incubation of the bacteria with the peptides resulted in a substantial cytotoxic effect (Fig 4). However, the effect of the peptides was not as substantial as the commercially available antibiotic kanamycin. Pep5 continuously reduced bacterial viability even at 50 μM; 32. 5 ± 4.3% and up to 45.3 ± 4.3% reduction at a concentration of 125 μM. At a 100 μM concentration some peptides exhibited a certain level of bacterial viability reduction observed even without incorporating Cu(II) into the growth media. However, at 125 μM all peptides exhibited an effect independent of the presence of Cu(II). These results indicated that *E. coli* viability might be affected by those peptides even at the low copper concentration that naturally exists in growth media. Pep8 also exhibited a substantial reduction of bacterial viability though not as much as pep5; the largest reduction of bacterial viability was detected at a concentration of 125μM: 41 ±5.15%.

In order to verify that these peptides are specifically toxic to microbes and not to mammalian cells, we carried out toxicity experiments (Fig M in S1 File), which showed that even at 150 μM, highly differentiated rat L6 myotubes remain alive.

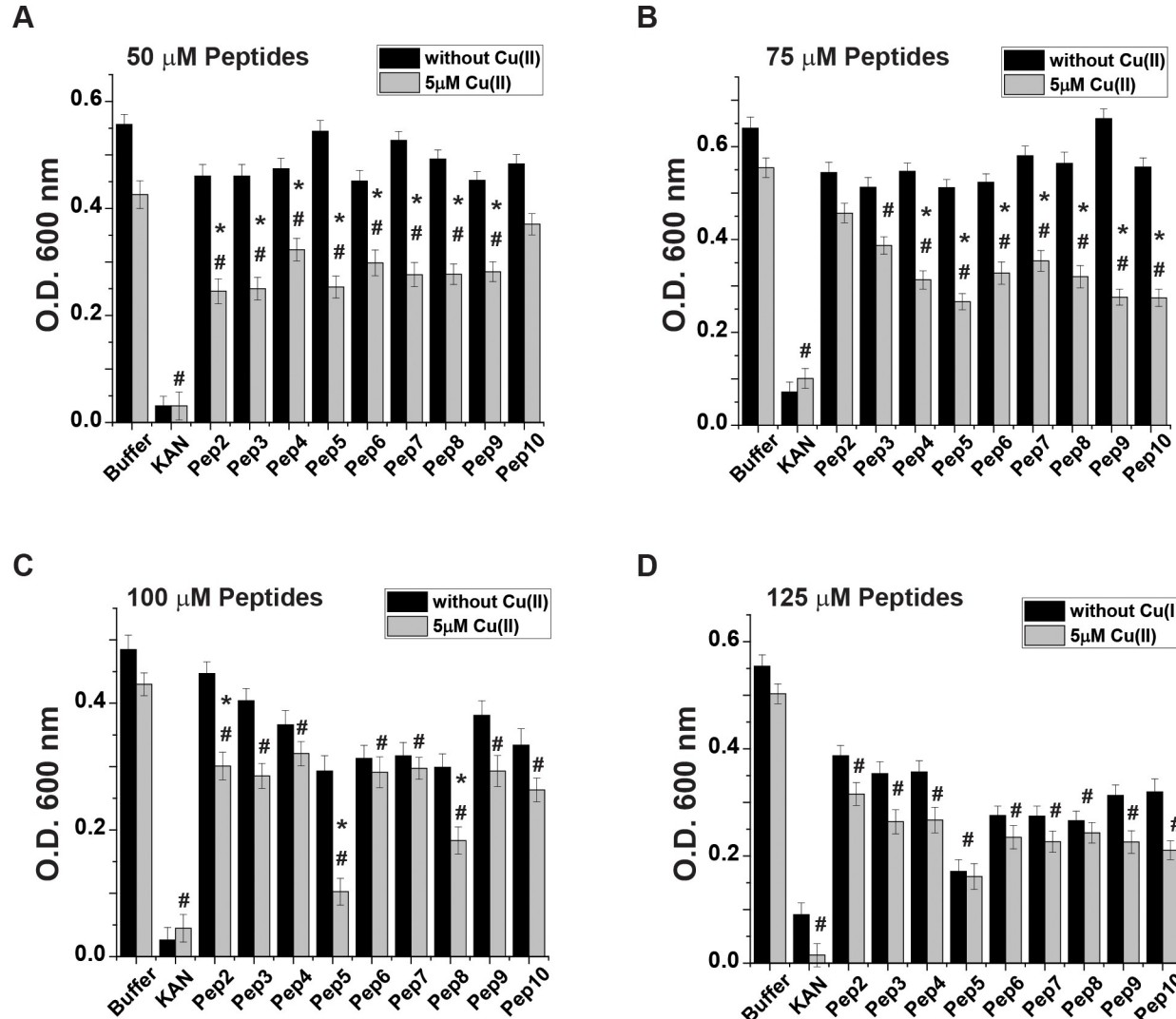

**Fig 3. Effect of test peptides on the *E. coli* growth rate.** *E. coli* cultures were grown as described in Methods. The growth rate values after a 14 h incubation in the medium in the presence (gray lines) and absence (black lines) of Cu(II). Concentrations of the test peptides: A. 50 μM, B. 75 μM, C. 100 μM, and D. 125 μM. $^*p < 0.05$, MEAN ± SE, n = 5, a comparison between the values in the presence and absence of Cu(II). #$p < 0.05$, MEAN ± SE, n = 5, in the presence of Cu(II) and comparing between values of a buffer and peptide. t-test was used for the statistic evaluation.

## Targeting conformational changes induced by pep5 and pep8 in the CusB structure by DEER measurements

For obtaining structural and dynamic information on CusB inhibition, double electron-electron resonance (DEER) spectroscopy, accompanied by site-directed spin labeling (SDSL), was used. DEER, also known as pulsed electron double resonance (PELDOR), is a pulsed electron paramagnetic resonance (EPR) method used to measure dipolar interactions between two or more electron spins. Thus, it can provide nanometer-scale interspin distance information in the range of 1.5–8.0 nm [45, 57–64]. Since proteins are diamagnetic species, nitroxide spin labels (paramagnetic) are attached to cysteine residues at selected positions within the protein, most commonly a nitroxide, 1-oxyl-2,2,5,5-tetramethylpyrroline-3-methyl methanethiosulfonate spin label (MTSSL) [65, 66]. The combination of DEER and SDSL previously allowed us to target conformational changes in CusB associated with Cu(I) binding [33]. CusB assumed a

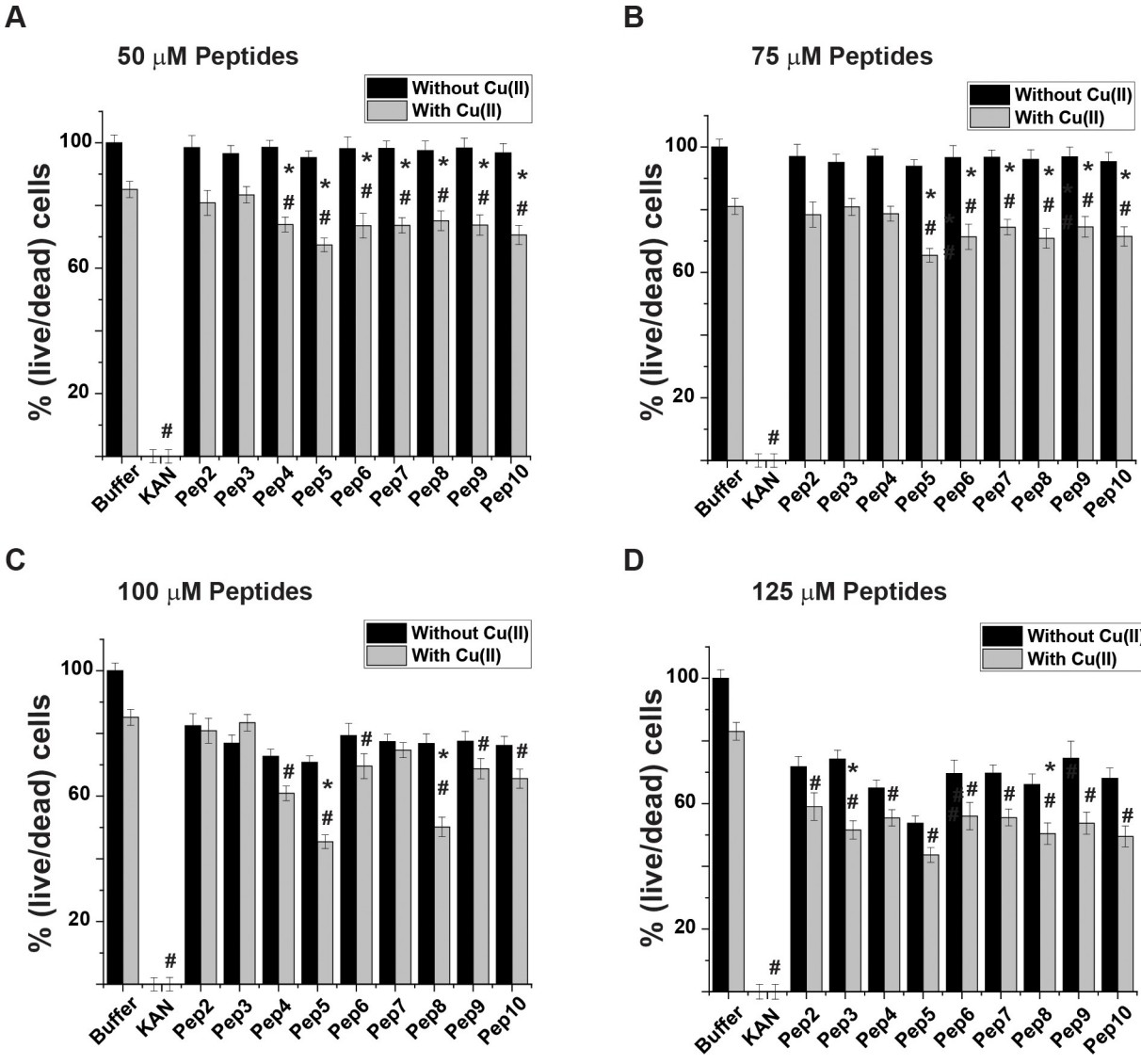

**Fig 4. Effect of the test peptides on *E. coli* viability.** *E. c*oli cultures were grown as described in Methods. After 14 h, test peptides were introduced to the medium in the presence (gray bars) and absence (black bars) of Cu(II). Concentrations of the test peptides: A. 50 μM, B. 75 μM, C. 100 μM, and D. 125 μM. *p < 0.05, MEAN ± SE, n = 3, a comparison between the values in the presence and absence of Cu(II). #p < 0.05, MEAN ± SE, n = 3, in the presence of Cu(II) and comparing between values of a buffer and peptide. t-test was used for the statistic evaluation.

more compact structure upon Cu(I) binding, whereas most of the structural changes occurred in domains 2 and 3 of CusB [33]. CusB lacks natural cysteine residues for labeling, so we mutated and spin-labeled CusB at A236C (Domain 3) and at A248C (Domain 2). Since CusB is a dimer in solution [33], the distance distribution between the four spin labels was used for the measurement (Fig 5). For apo-CusB (CusB without Cu(I) bound), distance distributions of 2.5 ± 0.5 nm were detected, whereas the holo-CusB (Cu(I)-bound CusB) showed distances of 2.8 ± 0.3 nm. These distance distributions are in agreement with the accepted distributions for the apo- and holo- CusB models.[48] When CusB was bound to pep5 and incubated with Cu (I) solution, a distance distribution similar to apo wt-CusB was observed, suggesting that the holo-CusB structure is inhibited by pep5. A similar phenome was obtained when M64 was

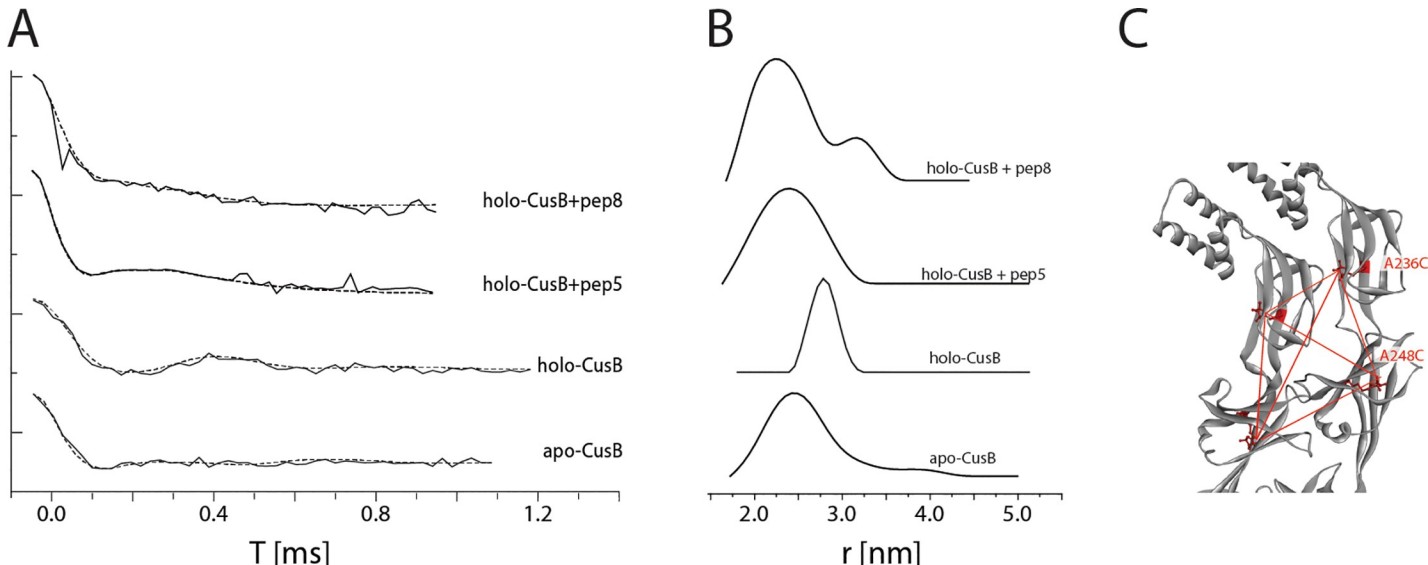

**Fig 5. Distance distribution measurements (DEER) on CusB.** A. DEER time domain signals (solid lines) and fits (dashed lines) based on Tikhanov regularization for apo-CusB, holo-CusB, and holo-CusB+pep5, CusB+pep8 (in the presence of Cu(I) at a Cu(I):CusB ratio of 3:1) spin labeled at A236C and A248C positions. B. Corresponding distance distribution functions. C. The positions of the spin labels in CusB dimer (A236C, A248C). The red dotted lines denote the six distances targeted here under the broad distance distribution.

mutated to isoleucine and no Cu(I) uptake by CusB occurred.[48] The similar distance distribution between holo CusB in the presence of pep5 and apo-CusB indicates that pep5 does not completely disrupt the dimerization of CusB, but instead, binds to CusB and induces mild structural changes without affecting the assembly of the whole CusCBA channel. However, the binding of pep5 to CusB inhibits structural changes in the presence of Cu(I) ions. When CusB was bound to pep8 and then incubated in Cu(I) solution, the distance distribution suggested

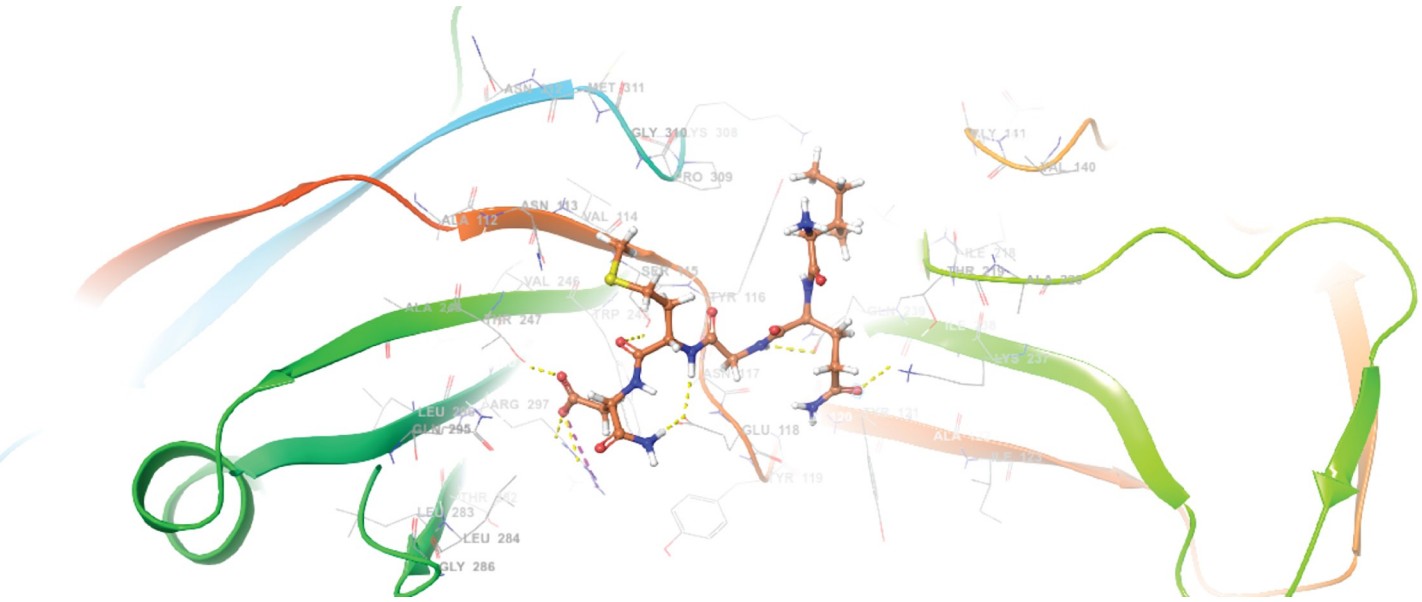

**Fig 6. The binding mode of docked pep 5 at the CusB binding site (PBD code: 3H94). B. 2D ligand interaction diagram between pep5 and CusB binding pocket.**

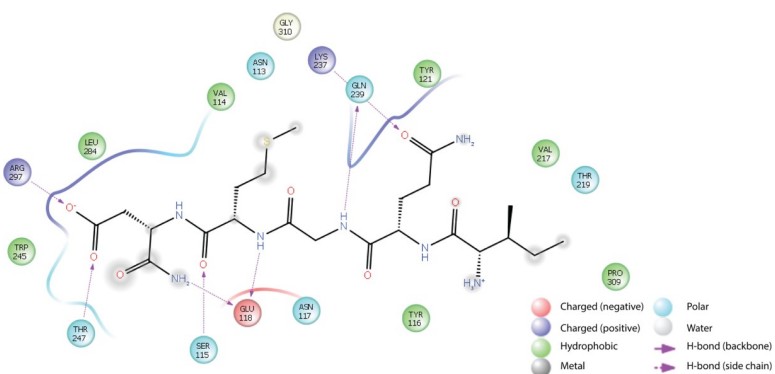

**Fig 7. 2D ligand interaction diagram between pep5 and CusB binding pocket.**

that the solution consists of a heterogeneous mixture of CusB protein that did not undergo conformational changes (a profile similar to apo-CusB), and that CusB underwent some structural changes (distribution around 3.2 nm).

Finally, the binding mode of the most active peptide (pep5) to a monomer CusB binding site was estimated *in silico* (Fig 6). Several important interactions were identified, as shown in Fig 7. The amide moiety of the C terminus of pep5 forms an H bond with Glu118 of CusB, and its side chain carbonyl interacts with Arg297 and Thr247. In addition, two -NH- moieties in the peptide backbone of pep5 binds to CusB *via* interaction with Glu118 and Gln239. Finally, Gln239 also is able to interact with the carbonyl in the side chain of pep5 (Gln).

## Discussion

The need for innovative antibiotics with novel bactericidal mechanisms is increasing rapidly. We propose an approach for developing novel antibiotics based on a unique molecular target: inhibition of the CusCFBA efflux system. The CusCFBA efflux system within Gram-negative bacteria plays a crucial role in copper homeostasis within the bacteria, which is directly associated with bacterial viability. We focused here on CusB, which represents a significant part of the efflux channel, and is responsible for initiating its opening and for exporting copper ions to the extracellular domain [29, 32]. In one of our previous studies, we found that CusB adopts a specific conformation in association with Cu(I) [33]. A physiologically active CusB domain is a trimer of core dimers. We determined that domains 2 and 3 (two central domains) of CusB undergoes major structural changes upon Cu(I) binding. More specifically, both domains are moved outside of the channel and are exposed to binding with another CusB monomer. In addition, based on crystallographic studies, the targeted region has been suggested to be highly flexible and therefore greatly influences the entire structure of the channel [24]. We hypothesized that in the presence of the small, CusB-like peptide, the formation of the full channel would not occur properly. By combining molecular modeling, magnetic resonance spectroscopy, and *in vitro* measurements, we have developed and tested a series of peptides, two of which (pep5 and pep8) exhibited a cytotoxic effect in *E. coli* culture in a Cu-dependent manner. Pep5 exhibited the greatest effect on cell growth and viability. Although we did not prove how and where pep5 interacts with CusB, we have substantial circumstantial evidence for this interaction. Pep5 was much more active in the presence of 5 μM Cu(II) in the growth media at all tested concentrations, indicating that the copper efflux system within the bacteria was affected by the peptide. Pulsed EPR spectroscopy was utilized to better validate this hypothesis, and the data suggest that pep5 targets the interference surface of the CusB

dimer by making structural changes, and that the native transformation between the apo to the holo state of CusB no longer exists in the presence of the peptide. We believe that this peptide does not affect the assembly of the whole transporter; however, in some way, and owing to its small size, it succeeds to penetrate between two monomers of CusB and thus, to inhibit physiological structural changes of the transporter upon copper stress. Accumulation of the copper most likely leads to oxidative stress *via* induced Fenton reaction, culminating in reduced bacterial viability. Since the system is unique to bacteria and is located across the periplasm of the bacteria, pep5 and future peptidomimetic derived compounds should be able to inhibit CusB activity by only penetrating a single membrane, making the target more accessible for inhibition.

Observing the large reduction in bacterial growth at a 125 μM concentration of pep5 even before the addition of copper suggests that at high peptide concentrations, the bacteria lose their ability to cooperate even with the minimal concentration of copper found in the bacterial growth media under physiological conditions. We could hypothetically make such a statement because at an identical concentration mammalian cells were not affected (Fig M in S1 File). Although the lowest effective concentration of pep5 is still a non-pharmacological concentration, we have conclusively shown that disruption of the Cu transport system in bacteria can lead to a cytotoxic effect, a discovery that should be given further study for its implications in treating antibiotic-resistant bacterial infections. We anticipate that pep5 can serve as a structural template for developing novel more potent and effective pep5 based peptidomimetics to disrupt CusB function more efficiently or even completely.

Such compounds may represent in the future a new class of antibiotics that can contribute to the fight against antibiotic resistance.

## Supporting information

**S1 File. Additional experimental data.**
(PDF)

## Author Contributions

**Conceptualization:** Aviv Meir, Lada Gevorkyan-Airapetov, Olga Viskind, Arie Gruzman, Sharon Ruthstein.

**Data curation:** Aviv Meir, Veronica Lepechkin-Zilbermintz, Fabian Schwerdtfeger, Anna Munder, Sharon Ruthstein.

**Formal analysis:** Aviv Meir, Veronica Lepechkin-Zilbermintz, Fabian Schwerdtfeger, Sharon Ruthstein.

**Funding acquisition:** Sharon Ruthstein.

**Investigation:** Aviv Meir, Shirin Kahremany, Fabian Schwerdtfeger, Lada Gevorkyan-Airapetov, Anna Munder, Olga Viskind, Sharon Ruthstein.

**Methodology:** Aviv Meir, Veronica Lepechkin-Zilbermintz, Shirin Kahremany, Lada Gevorkyan-Airapetov, Arie Gruzman, Sharon Ruthstein.

**Project administration:** Arie Gruzman.

**Resources:** Arie Gruzman, Sharon Ruthstein.

**Software:** Shirin Kahremany.

**Supervision:** Arie Gruzman, Sharon Ruthstein.

**Validation:** Shirin Kahremany, Sharon Ruthstein.

**Writing – original draft:** Aviv Meir, Arie Gruzman, Sharon Ruthstein.

**Writing – review & editing:** Arie Gruzman, Sharon Ruthstein.

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
