## [Decision Letter · Decision Letter 0]

12 Nov 2019

PONE-D-19-26397

Inhibiting the copper efflux system in microbes as a novel approach for developing antibiotics

PLOS ONE

Dear Prof. Ruthstein, dear Sharon

Thank you for submitting your manuscript to PLOS ONE. After careful consideration, we feel that it has merit but does not fully meet PLOS ONE’s publication criteria as it currently stands. Therefore, we invite you to submit a revised version of the manuscript that addresses the points raised during the review process.

As you will see, I was only able to secure one expert review in a reasonable time but the single review is very positive and is reinforced by my personal judgement of your study. Hence, we would recommend taking into account the points raised in the review before re-submission.

We would appreciate receiving your revised manuscript by Dec 27 2019 11:59PM. To enhance the reproducibility of your results, we recommend that if applicable you deposit your laboratory protocols in protocols.io, where a protocol can be assigned its own identifier (DOI) such that it can be cited independently in the future. For instructions see: http://journals.plos.org/plosone/s/submission-guidelines#loc-laboratory-protocols

We look forward to receiving your revised manuscript.

Kind regards,

Dariush Hinderberger

Academic Editor

PLOS ONE

Journal Requirements:

Reviewers' comments:

Reviewer's Responses to Questions

**Comments to the Author**

1. Is the manuscript technically sound, and do the data support the conclusions?

Reviewer #1: Yes

2. Has the statistical analysis been performed appropriately and rigorously? 

Reviewer #1: Yes

3. Have the authors made all data underlying the findings in their manuscript fully available?

Reviewer #1: Yes

4. Is the manuscript presented in an intelligible fashion and written in standard English?

Reviewer #1: Yes

5. Review Comments to the Author

Reviewer #1: Summary

The aim of this paper is to detail the design and effect of small peptides as inhibitors of the CusB protein – an important component of the bacterial copper efflux system. In the paper, the authors explain how these peptides were designed using computational modeling and the subsequent effect of these peptides on bacterial cell growth and viability. Through these experiments, the authors identify two peptides that have a significant effect on cell growth and viability. In addition, the authors utilized EPR DEER distance measurements to probe the structural changes of the CusB protein in the absence and presence of the most detrimental peptides. These distance measurements demonstrate how the holo form of the protein is not sampled in the presence of these peptides, thereby restricting the structural changes needed by CusB to facilitate Cu(I) transport from the cell.

I found this paper to be a thorough investigation of a peptide-based protein inhibition system. The results indicate a significant effect of peptide binding on CusB function in presence and even absence of Cu, thereby reducing not only the cell growth but viability of the cells. The EPR data provide essential structural information that helps to illustrate the effects of peptide binding on CusB structure.

I recommend this paper for acceptance after minor revisions.

Major issues

No major issues with this paper.

Minor issues

1. Line 212: BL21 cells not Bl21

2. Line 215: I do not understand the phrase “..with a script fully described next.” With respect to use of ImageJ software

3. Line 240: Please clarify what is meant by the 13-amino acid peptide containing too many features, I am guessing the authors are referring to features used in the Pharmacore program, but cannot be sure

4. Line 245: I do not see the red circle in Figure 1 indicating the region where the peptide overlaps with the binding site, please include or clarify in figure or caption.

5. I would suggest using the same orientation and color coding for the protein model in Figures 1 and 2 so the reader can more easily follow the orientation and geometry of the bound peptide

6. Lines 261-262: Authors refer to “All ten chose peptides…” being short sequences of the parent peptide (pep1) – in Table 1 there are a total of ten peptides, including the parent peptide. The sentence in lines 261-262 should then read “Nine chosen peptides were short subsequences of the parent peptide (pep1).”

7. Lines 262-263: Should start a new paragraph.

8. Line 263: “Several peptides were obtained in purity level, which is below 95%” should be re-written

9. Lines 264-265: Poorly written sentence at end of the paragraph.

10. Line 277: Define SOD, superoxide dismutase?

11. Line 279-280: Sentence about peptides remaining after incubation is poorly written

12. Line 376: C-terminus, singular not plural

13. Lines 419-420: Statement about peptide not affecting mammalian cells – reference to SI missing?

6. PLOS authors have the option to publish the peer review history of their article (what does this mean?). If published, this will include your full peer review and any attached files.

Reviewer #1: No

---

## [Author Response · Author response to Decision Letter 0]

19 Nov 2019

1. Line 212: BL21 cells not Bl21

Author replay: Corrected

2. Line 215: I do not understand the phrase “..with a script fully described next.” With respect to use of ImageJ software

Author replay: thanks, we deleted this sentence

3. Line 240: Please clarify what is meant by the 13-amino acid peptide containing too many features, I am guessing the authors are referring to features used in the Pharmacore program, but cannot be sure

Author replay: Thank you for the comment, we referred to too many pharmacophore features and added this information to the text in page 9 for clarity. 

4. Line 245: I do not see the red circle in Figure 1 indicating the region where the peptide overlaps with the binding site, please include or clarify in figure or caption.

Author replay: Thank you for the comment, we added the missing red circle and also changed the orientation and color coding as suggested in comment No.5.

5. I would suggest using the same orientation and color coding for the protein model in Figures 1 and 2 so the reader can more easily follow the orientation and geometry of the bound peptide

Author replay: Thank you for the comment, we addressed this in the above comment. 

6. Lines 261-262: Authors refer to “All ten chose peptides…” being short sequences of the parent peptide (pep1) – in Table 1 there are a total of ten peptides, including the parent peptide. The sentence in lines 261-262 should then read “Nine chosen peptides were short subsequences of the parent peptide (pep1).”

Author replay: we corrected it according to the reviewer’s suggestion

7. Lines 262-263: Should start a new paragraph.

Author replay: done

8. Line 263: “Several peptides were obtained in purity level, which is below 95%” should be re-written

Author replay: we rewrote and clarified this paragraph.

9. Lines 264-265: Poorly written sentence at end of the paragraph.

Author replay: corrected

10. Line 277: Define SOD, superoxide dismutase?

Author replay: we added the definition

11. Line 279-280: Sentence about peptides remaining after incubation is poorly written

Author replay: we corrected it

12. Line 376: C-terminus, singular not plural

Author replay: thanks - corrected

13. Lines 419-420: Statement about peptide not affecting mammalian cells – reference to SI missing?

 Author replay: yes, sorry for that, we added the reference to the S1 file.

---

## [Editor Report · Decision Letter 1]

12 Dec 2019

Inhibiting the copper efflux system in microbes as a novel approach for developing antibiotics

PONE-D-19-26397R1

Dear Dr. Ruthstein, dear SHaron

We are pleased to inform you that your manuscript has been judged scientifically suitable for publication and will be formally accepted for publication once it complies with all outstanding technical requirements.

With kind regards,

Dariush Hinderberger

Academic Editor

PLOS ONE
---

## [Editor Report · Acceptance letter]

16 Dec 2019

PONE-D-19-26397R1 

Inhibiting the copper efflux system in microbes as a novel approach for developing antibiotics 

Dear Dr. Ruthstein:

I am pleased to inform you that your manuscript has been deemed suitable for publication in PLOS ONE. Congratulations! Your manuscript is now with our production department. 

With kind regards,

on behalf of

Professor Dr. Dariush Hinderberger 

Academic Editor

PLOS ONE